# IL-17 cytokines preferentially act on naïve CD4+ T cells with the IL-17AF heterodimer inducing the greatest functional changes

**Michael P. Crawford**[1,2‡], **Nicholas Borcherding**[1‡], **Nitin J. Karandikar**[1,2]*

**1** Department of Pathology, University of Iowa Health Care, Iowa City, Iowa, United States of America,
**2** Iowa City Veterans Administration Medical Center, Iowa City, IA, United States of America

‡ MPC and NB contributed equally to this work as co-first authors.
* nitin-karandikar@uiowa.edu

## Abstract

CD4+ T-helper 17 (Th17) T cells are a key population in protective immunity during infection and in self-tolerance/autoimmunity. Through the secretion of IL-17, Th17 cells act in promotion of inflammation and are thus a major potential therapeutic target in autoimmune disorders. Recent reports have brought to light that the IL-17 family cytokines, IL-17A, IL-17F and IL-17AF, can directly act on CD4+ T-cells, both in murine and human systems, inducing functional changes in these cells. Here we show that this action is preferentially targeted toward naïve, but not memory, CD4+ T-cells. Naïve cells showed transcriptome changes as early as 48 hours post-IL-17 exposure, whereas memory cells remained unaffected as late as 7 days. These functional differences occurred despite similar IL-17 receptor expression on these subsets and were maintained in co-culture/transwell systems, with each subset maintaining its functional response to IL-17. Importantly, there were differences in downstream transcriptional signaling by the three IL-17 cytokines, with the IL-17AF heterodimer conferring both the greatest transcriptional change and most altered functional consequences. Detailed transcriptome analysis provides important insights into the genes and pathways that are modulated as a result of IL-17-mediated signaling and may serve as targets of future therapies.

## Introduction

The dysfunction of CD4+ T-helper 17 (Th17) cells has been associated with a number of autoimmune conditions, such as type 1 diabetes, multiple sclerosis, and arthritis [1–7]. As a major source of proinflammatory IL-17 secretion, Th17 cells can act on stromal cells to produce inflammatory cascades and recruit neutrophils [8]. In addition, the differentiation of Th17 cells has a bifurcated relationship with the differentiation of regulatory T-cells (Tregs), a suppressive T-cell subset, in the periphery [9]. In the current paradigm, the signaling downstream of TGF-β in a naïve CD4+ T-cell can lead to Th17 or Treg differentiation. However, in the presence of proinflammatory cytokines, the fate is biased toward Th17 differentiation. These

**Data Availability Statement:** All data are included in this article, with the exception of raw RNAseq data, which can be accessed in the NCBI Genome

Expression Omnibus public database through GEO Series accession no. GSE150805.

**Funding:** This work was supported, in part, by grant awards from the NIH and VA to N.J.K. (R01 AI121567, I01 CX002319) and N.B. (F30 CA29655). The funders had no role in study design, data collection and analysis, decision to publish, or preparation of the manuscript.

**Competing interests:** The authors have declared that no competing interests exist.

observations have led to the evaluation of anti-IL-17 therapies for inflammatory and autoimmune conditions that have had mixed success.

We and others recently found that IL-17 is capable of acting directly on CD4⁺ T-cells in both humans and mice [10, 11]. We found that in the presence of a Th17-differentiating milieu, CD4⁺ T-cells become more resistant to CD8-mediated immune suppression [11], partly mediated by their production of IL-17A, IL-17F, and IL-17AF heterodimer. Moreover, these cytokines could also act directly on non-Th17 CD4⁺ T-cells and make them more resistant to suppression. This suppressive resistance could be ameliorated by interfering with IL-1β, IL-6 or STAT3 signaling [11]. Recently published studies in an autoimmune uveitis model also suggest a complex signaling network in developing Th17 cells, where feedback of IL-17A leads to NF-κb activation and subsequent secretion of IL-24 to suppress the Th17 genetic program [10]. Interestingly, the removal of IL-17A led to compensatory increases in IL-17F and GM-CSF. However, the knockout of IL-17F did not have the same effect, suggesting both redundant and unique signaling for IL-17 family members [10].

Here we further our investigation of IL-17-mediated changes in CD4⁺ T-cells, finding naïve, but not memory, CD4⁺ T-cells as the principal cells responding to IL-17. Characterizing the genetic patterns resulting from IL-17A, IL-17F and IL-17AF actions, we found that IL-17AF induced the greatest magnitude of suppressive resistance and size of genetic programs. We found general downregulation of cytokine and immune effector molecules and upregulation in downstream mediators of interferon signaling.

## Materials and methods

### Cell preparation and bead sorting

Peripheral blood mononuclear cells (PBMC) from healthy subjects were isolated from de-identified leukocyte reduction system (LRS) cones containing leukocyte-rich whole blood from platelet donors at the University of Iowa, DeGowin Blood Center. PBMC isolation was performed with BD Vacutainer CPT tubes (BD, 362753) density gradient centrifugation. CD8 T-cells were positively selected from freshly prepared PBMC with Manual LS Column MACS sorting with Miltenyi Biotec MACS CD8 Bead sorting microbeads (130-045-201) according to manufacturer specifications. Untouched bulk CD4⁺CD25⁻ T-cells were obtained using the CD4⁺ T-cell Isolation Kit (130-096-533) followed by a CD25 microbead depletion (130-092-983). CD45RO (130-046-001) or CD45RA (130-045-901) positive selection microbeads were used to respectively obtain CD4⁺CD25⁻CD45RO⁺ memory T-cells and "untouched" CD4⁺CD25⁻CD45RO⁻ naïve T-cells or CD4⁺CD25⁻CD45RA⁺ naïve T-cells and "untouched" CD4⁺CD25⁻CD45RA⁻ memory T-cells. Sort purities of all populations were routinely above 93% by flow cytometric analysis (S1 Fig). Sorted CD8 T-cells and CD4⁺ T-cell populations were frozen in human serum and dimethyl sulfoxide-containing media on the day of sorting for future use.

### Th subset differentiation

Memory CD4⁺ T-cells were subjected to Th0, Th1, Th2 and Th17 differentiation cultures, as described previously [11]. Briefly, memory cells were thawed in RPMI 1640 (Corning 10-040-CV) with DNase at 10KU/ml (Sigma D4513-1vl) and then resuspended at $1 \times 10^6$ cells/ml in X-VIVO 15 serum-free media (Lonza, 04-418Q), followed by stimulation in various differentiation conditions (Media Alone/Th0, Th1, Th2, Th17). Conditions included: (1) Media Alone/Th0: no cytokines/antibodies added; (2) Th1: anti-IL-4 7μg/ml BD554481, IL-2 10ng/ml BD554603, IL-12 10ng/ml BD554613; (3) Th2: anti-IFNγ 7μg/ml BD554698, IL-2 10ng/ml, IL-4 10ng/ml BD554605; (4) Th17: anti-IL-4 7μg/ml, anti-IFNγ 7μg/ml, TGFβ1 10ng/ml

eBiosciences 14-8348-62, IL-1β 10ng/ml BD554602, IL-6 50ng/ml BD550071. Cultures were activated with 1 μg/ml each of fixed anti-CD3 (eBiosciences, 16-0037-85) and anti-CD28 (eBiosciences, 16-0289-85) and incubated for 7 days at 37˚C, followed by suppression assay cultures. n = 7

## IL-17 receptor staining

IL17RA APC (Miltenyi Biotec #130-104-722), REA-APC isotype control (Miltenyi Biotec #130-104-614), IL17RC PE (Miltenyi Biotec #130-109-150) and the REA-PE isotype control (Miltenyi Biotec #130-104-612) were used to stain ex vivo cells and T-helper subsets. Flow cytometric data was acquired on the Cytek Aurora flow cytometer and analyzed with FlowJo software v10.

## Transwell exogenous IL-17 addition suppression assays

The 24 multiwell tissue culture plates (Costar Ref 3413, tissue culture treated polystyrene with 6.55 mm insert and 0.4 micrometer polycarbonate membranes) were plated with 1 μg/ml each of anti-CD3 and anti-CD28, as previously described [12]. Anti-CD3/anti-CD28 1ug/ml was fixed per manufacturer protocol to Sperotech polystyrene protein G particles (cat PGP-60-05, 0.5 w/v, 6.8 um) and were used within the well insert for stimulation. *Ex vivo* bead-sorted memory and naïve CD4+ T-cell populations were cultured in X-VIVO 15 media. IL-17A (eBioscience, 34-8179-82), IL-17F (R&D Systems, 1335-IL-025/CF), IL-17A+IL-17F and IL-17AF (R&D Systems, 5194-IL-025/CF) were added at 10 ng/mL to the indicated cultures (Fig 3) and cultured for 7 days. At 7 days, the cells were removed from the transwell, washed and used as responder cells in standardized suppression assays.

## Flow cytometric suppression assays

Either ex vivo- or 7-day-cultured CD4+ T-cells were placed in flow cytometric suppression assays, as described previously [11, 13, 14]. Briefly, responder CD4+ T-cells were stained with CFSE, followed by culture 1 μg/ml of fixed anti-CD3 (eBiosciences, 16-0037-85) and 1μg/ml of fixed anti-CD28 (eBiosciences, 16-0289-85) in the presence or absence of *ex vivo* sorted autologous bulk CD8+ T-cells at 1:0, 1:1 and 1:0.5 CD4:CD8 cell concentrations. On day 7 of culture, cells were stained for anti-CD4 PE-Cy7 (BD, 557852), anti-CD3 AlexaFluor700 (BD, 557943), anti-CD8 Pacific Blue (Biolegend, 344718) and flow cytometrically assessed for CD4 proliferating fraction (CFSE dilution). Percent proliferation and percent suppression were calculated as described previously [13]. Other antibodies utilized include: CD45RO Pacific Blue, Biolegend 304216; CD45RO PE, Biolegend 304244; CD45RA Fitc, BD 555488; CD8 PE, BD 340046; CD8 BV786, BD 563823; CD4 BV786, BD 563877; CD4 APC, BD 561841; CD4 PE, BD 555347; CD3 APC, Biolegend 300412; CD25 PE, BD 555432; CD25 APC, BD 555434; CD25 PacBlue, Biolegend 356130. Cells were analyzed on a BD LSRII or Cytek Aurora. Data were analyzed with FlowJo software v10.

## RNA sequencing/transcriptome analysis

Ex vivo-purified bulk CD4+CD25- T-cells and memory CD4+CD25-CD45RO+ T-cells were activated in vitro for 48 hours or 7 days in the presence of media alone (controls), or 10 ng/ml of IL-17A (eBioscience, 34-8179-82), IL-17F (R&DSystems, 1335-IL-025/CF) or IL-17AF (R&D Systems, 5194-IL-025/CF). Cells were incubated with the respective cytokines 90 minutes prior to activation and culture. Supernatants were separated and frozen for later analysis. Cell pellet samples were snap frozen and were submitted to the University of Chicago

Genomics facility for RNA extraction, quality assessment and sequencing. Single-end 50 bp sequencing was performed on the Illumina HiSeq 2000 (Illumina, San Diego, CA). Pseudoalignment was performed using kallisto with the GRCh38 human genome build [15]. Pseudoalignments were processed using sleuth (v0.30) R package using gene-level quantifications [15]. Differential gene expression analysis was performed in the sleuth R package with the Wald test. Differential genes were defined as log2-fold change >1 or <-1 and false discovery rate < 0.05. The significant genes were used for Ingenuity Pathway Analysis (Qiagen) using the same cut points for significance as inputs. Overlap coefficients were calculated using the size of the intersection of the conditions divided by the total size of the smallest condition.

### ELISAs

ELISAs were performed per manufacturer protocol on culture supernatant aliquots of original RNAsequenced cells. Human ELISA Kits for IL6 (Invitrogen, BMS213-2), IL-1β (Invitrogen, KAC1211), IL-10 (Invitrogen, BMS215/2), CXCL1 (Invitrogen BMS2122), CCL1/I-309 (Invitrogen, EHCCL1), IL-4 (Invitrogen, BMS225-2), IL-5 (R&D D5000B). ELISA data were acquired on a BioTek Synergy H1 Hybrid Reader. Gen5 v2.09 was used for software analysis.

### Statistics

For non-RNAseq data, Graphpad Prism v7.03 was used for statistical graphics using paired $t$ tests for significance.

### Study approval

All experiments were performed on PBMC obtained from de-identified leukocyte reduction system (LRS) cones from healthy platelet donors at the University of Iowa DeGowin Blood Center, as approved by the University of Iowa Institutional Review Board.

## Results

### IL-17 cytokines affect naïve and bulk CD4+ T-cells but show no effect on memory T-cells

For these studies, we used Miltenyi magnetic bead cell sorting to yield total (bulk) CD4+ CD25- T-cells from healthy donors and further isolated CD45RO+ memory T-cells (Fig 1A), with a sort purity/enrichment of 93.62% +/- 1.13. The bulk *ex vivo* CD4+25- T cell populations were made up of ~57% memory and ~42% naïve T-cells (S2 Fig). Cells were first incubated with the indicated cytokines and then activated with anti-CD3 and anti-CD28. We expected to see a large change in memory CD4+ T-cells due to their inherently lower threshold for activation. However, we found significantly greater differential gene expression in the bulk CD4+ T-cells at 2 days of activation (Fig 1B). Therefore, we activated memory CD4+ T-cells for 7 days and yet did not observe virtually any changes in the number of differentially expressed genes (Fig 1B). This indicated that the naïve T-cell component of the bulk population may be responsible for IL-17-induced resistance to suppression. To test this functionally, we isolated CD4+CD25- bulk, CD4+CD25-CD45RA+RO- naïve, and CD4+CD25-CD45RA-RO+ memory T-cells. To ascertain that the functional effects were not an artifact of magnetic microbeads stuck to the positively sorted cells, we performed the sorting using CD45RA or CD45RO microbeads, resulting in positive or negative selection of each subset, leaving the other subset "untouched." These cells were activated in media or IL-17AF for 7 days and then assessed for their suppressive resistance, as described previously [11]. Regardless of the beads used

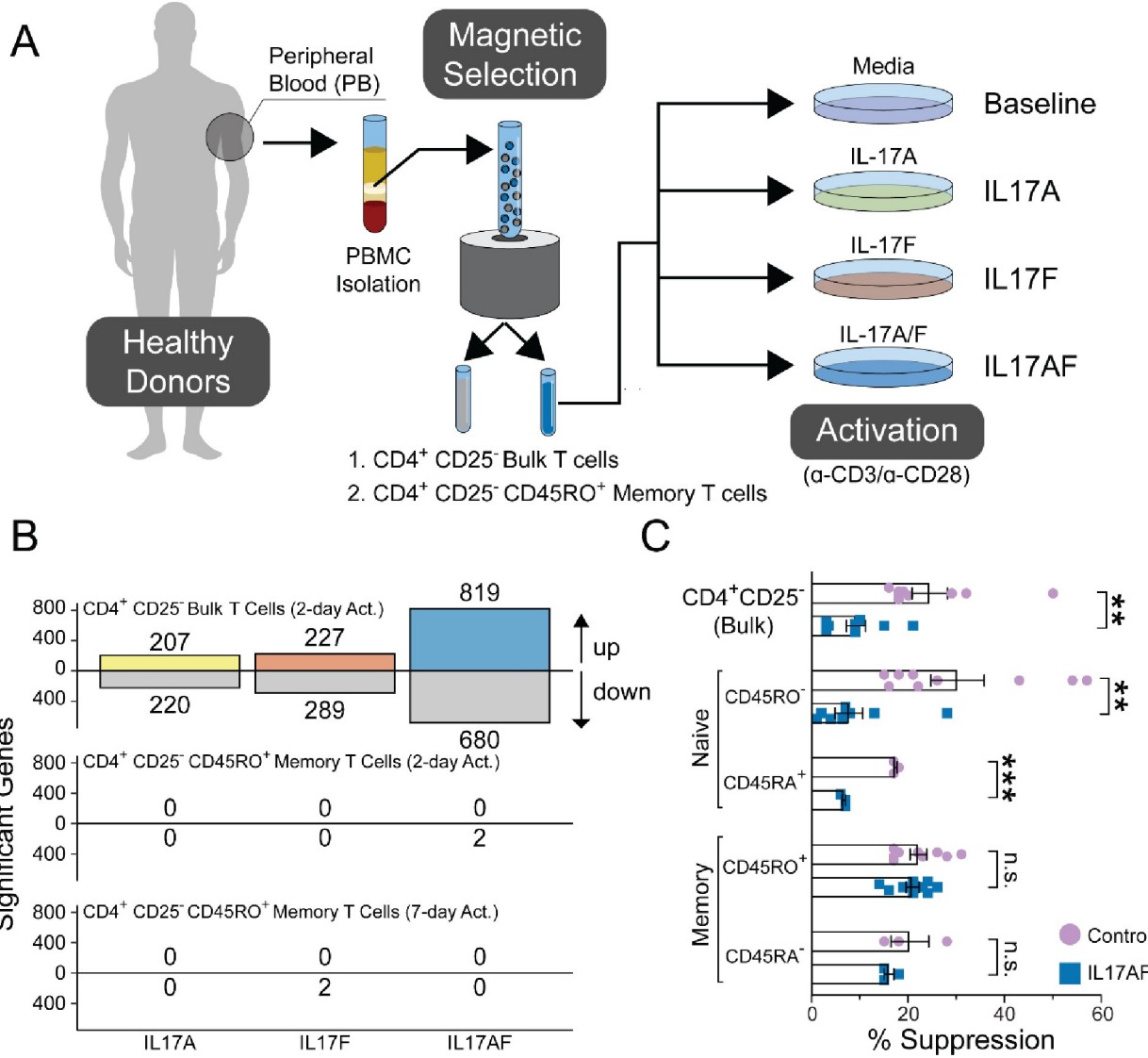

**Fig 1. IL-17 cytokines affect naïve and bulk CD4+ T-cells but show no effect on memory T-cells. A**. Schematic summary of the *ex vivo* treatment of bead-sorted bulk or memory CD4+ CD25- T-cells activated in media alone (control/baseline), IL-17A, IL-17F, or IL-17AF heterodimer. After activation, RNA was isolated and underwent mRNA sequencing. **B.** The number of significant genes upregulated and downregulated with IL-17A (yellow), IL-17F (orange) and IL-17AF (blue) within the indicated populations. Significance was defined as q-value < 0.05 and log2-fold change > 1 or < -1. **C.** Negatively selected CD4+25- T-cells were further sorted using either CD45RO or CD45RA magnetic beads resulting in bulk, naïve and memory T-cell populations, which were αCD3/αCD28-activated for 7 days in media alone (control) or in the presence of IL-17AF and then used as responder cells in flow cytometric suppression assays with autologous CD8+ T-cells as suppressors. ** p<0.005, *** p<0.001, n.s. = not significant by Paired Student T-test.

(CD45RO or CD45RA) or the positive/negative isolation (bead touched/untouched) status, we found that bulk and naïve CD4+ T-cells attained significant suppressive resistance following IL-17 exposure (Fig 1C). The memory cells were unaffected by this exposure, functionally corroborating the transcriptome data. Thus, IL-17 cytokines consistently induced functional changes in the naïve CD4+ T-cell subset but did not affect memory CD4+ T-cells.

## Memory and naïve CD4⁺ T-cells have similar levels of IL-17 receptor expression, are similarly suppressible and have similar plasticity and resistance patterns when exposed to Th0/Th1/Th17 differentiation conditions

Next, we investigated the possibility that the IL-17 receptors were differentially expressed on memory versus naïve CD4⁺ T-cells. Fig 2A shows that IL17RA and IL17RC receptor expression patterns were not significantly different between these populations, with >80% of cells expressing IL17RA and <20% of cells expressing IL17RC. Similarly, previously activated and differentiated Th0 and Th17 cells also showed similar proportions of IL17RA- and IL17RC-expressing cells (Fig 2B and 2C). Moreover, CD8 T-cell-mediated suppression of *ex vivo* memory CD4⁺ T-cells was similar to *ex vivo* naïve CD4⁺ T-cells (Fig 2D). Notably, we did not observe any differences in % proliferation between *ex vivo* naïve vs memory in the 1:0, 1:0.5 or 1:1 culture conditions (Fig 2D; p = n.s.).

We have shown previously that naïve CD4⁺ T-cells cultured under different cytokine milieus attain differential resistance to suppression [11], with greatest resistance by Th17 cells. Interestingly, in these types of differentiation conditions, memory cells also showed functional plasticity in that their suppressive resistance was modulated in a manner similar to that of naïve CD4⁺ T-cells, including suppressive resistance following exposure to Th17 conditions

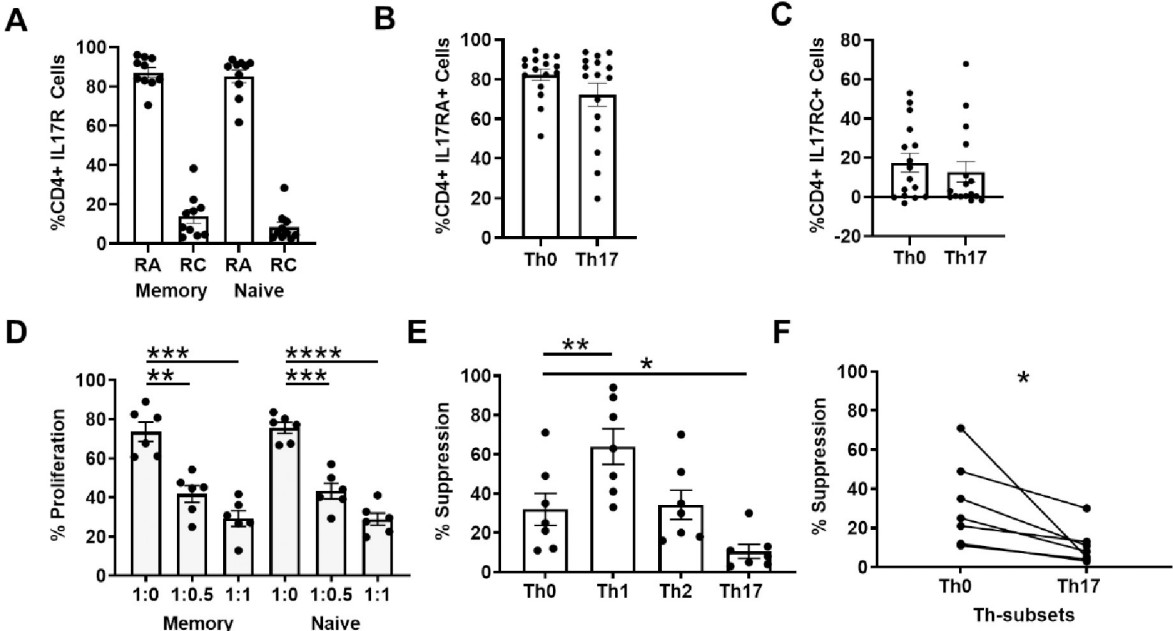

**Fig 2. IL-17RA/RC receptor expression does not account for the lack of memory cell response to IL-17. A.** IL17RA and IL17RC expression on *ex vivo* microbead-sorted CD4⁺ Naïve (CD45RA⁺ and CD45RO⁻) and CD4⁺ Memory (CD45RO⁺ and CD45RA⁻) T-cells (Mean+/-SEM; n = 9). **B-C.** IL17RA and IL17RC expression after 7-day differentiation cultures of Naïve CD4⁺ T cells into Th0 and Th17 conditions (Mean+/-SEM; n = 16). **D.** Ex vivo-purified memory (CD4⁺CD25⁻CD45RO⁺) and naïve (CD4⁺CD25⁻CD45RO⁻) T-cells from healthy donors were stained with CFSE, followed by a 7-day culture with autologous irradiated APCs, fixed αCD3 antibody, and +/-increasing numbers of CD8⁺ T-cells (at the indicated ratios). Column bars depict %proliferation (+/-SEM) of indicated CD4⁺ T-cells at the indicated CD4:CD8 ratios, showing similar proliferation at 1:0 and similar suppression at 1:0.5 and 1:1. **p<0.005, *** p<0.001, ****p<0.0001 by paired student t test; n = 6. **E-F.** *Ex vivo*-purified memory CD4⁺CD25⁻CD45RO⁺ T-cells from healthy donors were polarized under indicated differentiation conditions for 7 days and then placed in a 7-day CD8⁺ T-cell mediated suppression assay. (E) Column bars depict %suppression (+/-SEM) of indicated Th-subsets by CD8⁺ T-cells at the 1:0.5 CD4:CD8 ratio, showing greater sensitivity of "Th1 milieu"-exposed cells and greater resistance by "Th17 milieu"-exposed cells. (F) Paired %suppression data from the Th0 and Th17 suppression cultures. *p<0.05, ** p<0.005.

(Fig 2E and 2F). Thus, other cytokines could functionally modulate the suppressive resistance of memory CD4+ T-cells. However, unlike these other differentiation cytokines, IL-17 seems to preferentially act on naïve T-cells while uniquely ignoring *ex vivo* memory cells.

## Cytokines from IL-17-responsive naïve T-cells do not affect the IL-17-non-responsiveness of memory T-cells

Our initial observations (Fig 1) were made on bulk vs. memory CD4+ T-cells. We wondered whether the interactions of naïve and memory cells from bulk cultures might convert memory T-cells into becoming IL-17-responsive. For this, we used transwell assay systems where memory cells would get exposed to cytokines secreted by naïve T-cells in the presence of IL-17 (Fig 3). Thus, *ex vivo* CD4+25- T-cells were sorted into memory and naïve subsets and activated with anti-CD3 and anti-CD28 for 7 days in the presence or absence of IL-17A, IL-17F, IL-17AF or IL-17A+IL-17F either across from each other through a transwell (Fig 3A) or memory alone (Fig 3B) and naïve alone (Fig 3C). The subsets were then resuspended and placed in a CD8 T-cell mediated suppression assay. Naïve cells retained the IL-17-induced resistance to suppression whether cultured alone or in the presence of memory cells. In contrast, memory T-cells remained unresponsive to IL-17 in either condition, suggesting that they were not affected by secreted molecules from neighboring naïve cells. There were no significant differences between the suppression of memory cells that were cultured with naïve cells (through the transwell) versus those cultured alone (p = n.s.). Naïve cells cultured in transwells versus alone also showed largely similar degrees of suppressive resistance in most of the conditions, except for the IL-17AF condition, which showed significantly greater resistance when naïve cells were cultured alone, likely suggesting undiluted effect of the heterodimer. Overall, these results emphasize the distinct responsiveness of memory vs. naïve cells to IL-17 cytokines.

## Overlapping as well as unique transcriptome changes in IL-17-treated CD4+ T-cells

In order to better characterize the IL-17-mediated actions on CD4+ T-cells, we next examined the underlying genetic patterns from the three IL-17 conditions. We found that IL-17A and IL-17F treatment had a notable overlap in upregulated (overlap coefficient = 0.435) and downregulated (overlap coefficient = 0.536) genes compared with the IL-17AF heterodimer (Fig 4). Within upregulated genes shared in at least two conditions, we observed increases in interferon response factors (IRFs) and interferon-induced factors (IFITs) (Fig 4A and 4B). In contrast, commonly downregulated genes across the conditions included chemokines (*CCL2*, *CCL3*, and *CCL8*), cytokines (*IL1A*, *IL1B*, *IL6*, *IL17F*, and *IFNG*), and receptors (*CCR1* and *TLR8*) associated with T effector functions (Fig 4C and 4D). Of note, while the specific cytokine genes, *IL1B* and *IL6*, were downregulated in these conditions at the 48-hour snapshot, the alteration in these cytokines was the common component of the majority of differential pathways identified [11]. Moreover, interfering with the specific cytokines at a later timepoint during suppression assays led to the reversal of the suppressive resistance [11]. This shows that each of the potentially targetable genes or pathways would need to be dissected and studied longitudinally for specific functional effects.

Ranking the significant genes by log2-fold change, we found no clear difference in the distribution of the unique genes by incubation condition (Fig 4E). In addition, examining the top unique upregulated conditions, we noticed a general absence of immune-related genes for each condition (Fig 4F), with the exception of *IL21* in IL-17A and *IL36A* in IL-17AF.

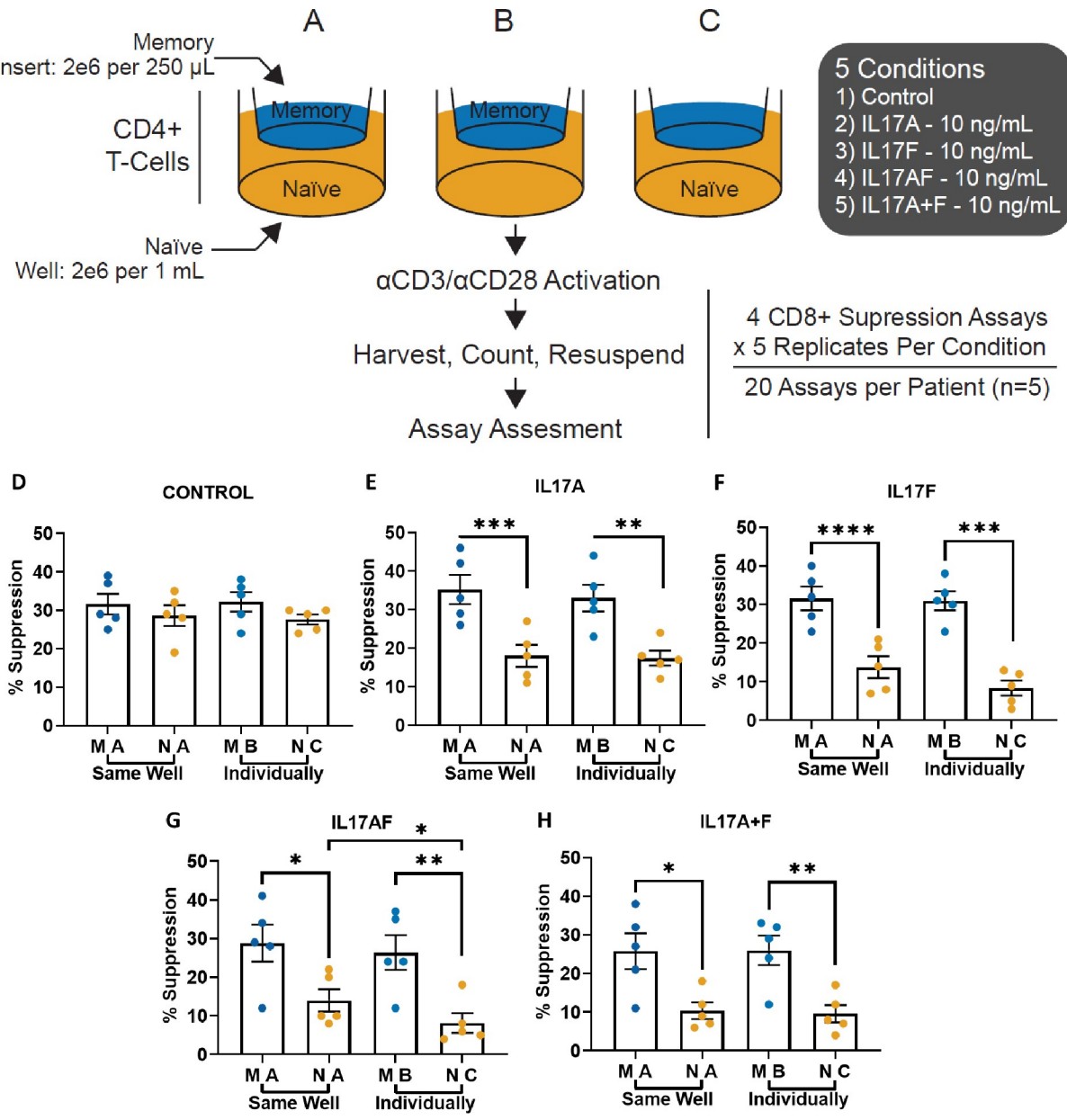

**Fig 3. Memory T-cell non-responsiveness to IL-17 is not affected by local auto/paracrine action of naïve T-cell cytokines.** Transwell cultures were set up in 24-well plates with transwell membrane inserts separating autologous presorted Memory and Naïve CD4+ T cells together (A) or individually (B and C), with indicated cytokine conditions (media alone, IL-17A, IL-17F, IL-17AF and IL-17A+IL-17F, panels D-H, respectively) stimulated with anti-CD3/anti-CD28-coated beads in culture for 7 days. On day 7, the cells were harvested, washed, stained with CFSE and subjected to CD8-mediated T-cell suppression assays with fixed anti-CD3 and anti-CD28 stimulation. Mean suppression +/- SEM for the 5 conditions is shown; n = 5 each; *p<0.05, **p<0.01, ***p<0.001, ****p<0.0001.

Using Ingenuity Pathway Analysis, we next examined the differential pathways for each condition compared to the baseline Th0. Across each condition, we noted a general trend towards increased enrichment of pathways and a high degree of overlap between IL-17A and IL-17F with an overlap coefficient of 0.60 (Fig 5A). Within common differentially altered pathways, we found increases in metabolic pathways and pathways downstream of cytokine signaling (Fig 5B), despite a general reduction in cytokine expression. Broad trends in common

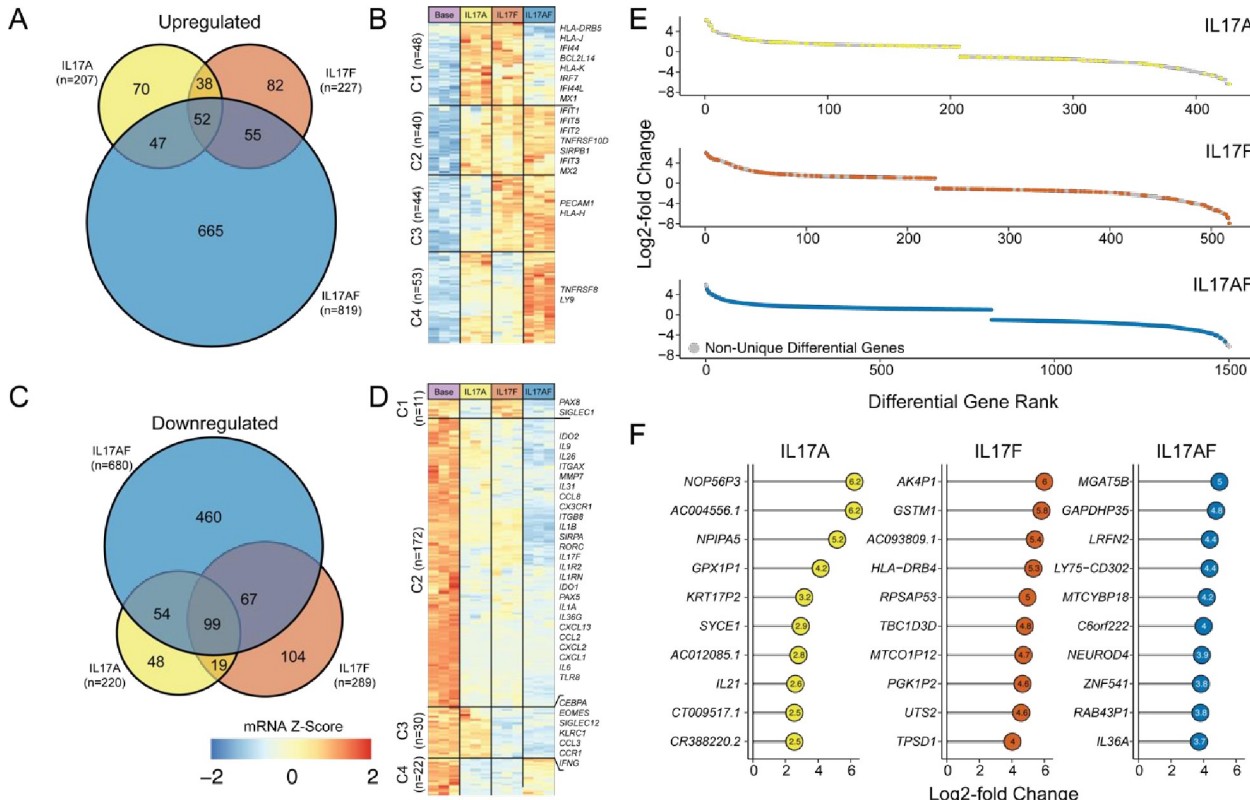

**Fig 4. Overlapping and unique expression patterns in IL-17-treated conditions.** Overlapping and unique upregulated genes in bulk CD4+CD25- T-cells in IL-17A, IL-17F, and IL-17AF conditions compared to media-only cells. **B**. Heatmap of upregulated genes in at least two of three conditions; clustering based on Euclidean distance with immune-related genes labeled. **C**. Overlapping and unique downregulated genes in bulk CD4+CD25- T-cells in IL-17A, IL-17F, and IL-17AF conditions compared to media-only cells. **D**. Heatmap of downregulated genes in at least two of three conditions; clustering based on Euclidean distance with immune-related genes labeled. **E**. Dispersion of differentially-regulated genes by condition, grey genes are common between at least two conditions and colored genes are unique to the indicated condition. **F**. Top 10 unique upregulated genes in each condition ordered by log2-fold change.

pathways with decreased enrichment were not observed, but IL-17F in Allergic Inflammatory Airway Disease, Toll-Like Receptor Signaling, and PPAR Signaling were seen (Fig 5B). Unique pathway enrichment by condition showed increased enrichment of acyl-CoA hydrolase and Cytotoxic T lymphocyte-mediated Apoptosis of Target Cells in IL-17A (Fig 5C). In IL-17F, top pathways enrichment, including PI3K signaling, Phospholipase C Signaling, and calcium-induced T lymphocyte Apoptosis (Fig 5C). The IL-17AF-treated cells had high levels of enrichment in EIF2 Signaling, a translation initiation factor, CREB Signaling and the Pyridoxal 5'phosphate Salvage pathway (Fig 5C), among a number of other pathways.

## Detailed analysis of significant IL-17AF-mediated changes in cytokines and chemokines

With IL-17AF leading to the greatest changes in gene expression, we wanted to further characterize this condition. We compared IL-17AF to the no cytokine baseline. We found a general decrease in chemokines (*CXCL1*, *CXCL3*, *CXCL5*, and *CXCL8*), activation markers (*LAG3* and *TNFRSF8*), effector molecules (*PRF1*), transcriptional regulators (*RORC*) (Fig 6). Taking a closer look at cytokines and chemokines (Fig 6B), we found an increase in a select number subsequent to IL-17AF treatment. These include interleukins (*IL4*, *IL5*, and *IL16*) and tumor

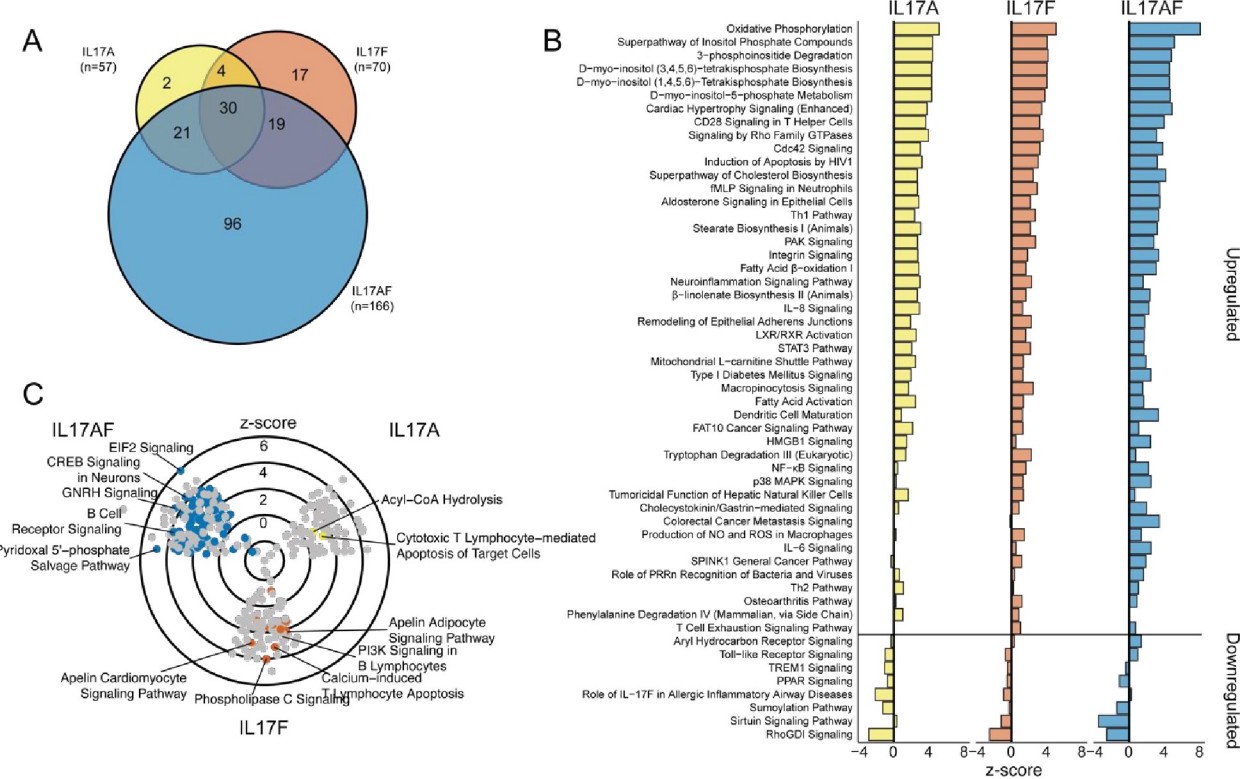

**Fig 5. Pathway induction following IL-17 exposure. A**. Overlapping and unique upregulated canonical pathway enrichment in bulk CD4⁺CD25⁻ T-cells in IL-17A, IL-17F, and IL-17AF conditions compared to media-only cells. **B**. Bar chart of significantly ($p < 0.05$) altered in at least two of three conditions. **C**. Z-score of significantly enriched canonical pathways with non-unique pathways colored in grey. Top increased unique pathways labeled.

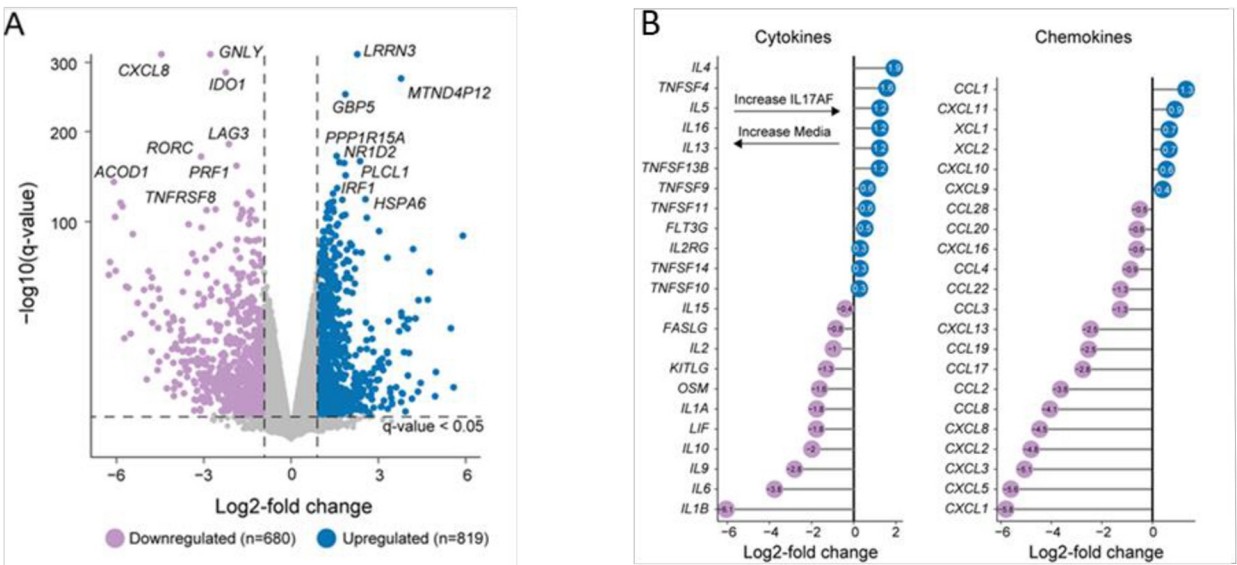

**Fig 6. Detailed analysis of significant IL-17AF mediated changes in cytokines and chemokines.** Volcano plot of differential genes identified comparing IL-17AF to media-only cells. Highlighted cells have q-value < 0.05 and log2-fold change > 1 or < -1. Selected genes labeled were in the top 20 genes by q-value. **B**. Filtered cytokine and chemokine genes with q-value < 0.05.

necrosis factor family of ligands (*TNFSF4*, *TNFSF9*, *TNFSF11*, and *TNFSF13B*), while the baseline condition had the highest levels of the *IL1B*, *IL6* and *IL10*.

To validate the RNAseq data, we selected some of the most up- and down-regulated cytokines/chemokines that were measurable by commercially available standardized ELISA assays. We performed these assays on supernatants from all four conditions (media, IL-17A, IL-17F and IL-17AF) on bulk cells at 48 hours, memory cells at 48 hours and memory cells at 7 days of culture. The production of IL-6, IL-1β, IL-10 and CXCL1 was indeed robust in the media-only condition for bulk CD4⁺ T-cells and significantly downregulated in the presence of IL-17, particularly in the IL-17AF condition (Fig 7). Conversely, IL-4, IL-5 and CCL1 were absent in the media condition and induced by IL-17AF, matching with the RNAseq analysis. Additionally, GM-CSF, IFNγ and sIL2r were measured as controls and not found to be different in these conditions (NCBI Genome Expression Omnibus public database Series accession no. GSE150805). Importantly, in contrast to bulk cells, the sorted memory cells did not show differences between media vs. IL-17 conditions at either 48h or 7d, corroborating the transcriptome data (Fig 1). Interestingly, memory cells did not produce appreciable quantities of IL-6, IL-1β or CXCL1 at baseline. Overall, these data validate the RNAseq analysis, demonstrating a strong CD4-intrinsic effect of IL-17, which is selective for naïve T-cells.

## Discussion

Our studies validate the recent observation that IL-17 cytokines can directly act on CD4⁺ T-cells and further demonstrate the unexpected finding that they act preferentially on naïve CD4⁺ T-cells while inducing no changes in memory CD4⁺ T-cells. One theoretical explanation of this differential effect on naïve versus memory CD4⁺ T-cells might be differences in IL-17 receptor expression. However, when we assessed two known receptors (IL-17RA and IL-17RC), we did not find significant differences in the ex vivo expression of these receptors between naïve vs. memory cells, either through flow cytometric staining (Fig 2) or by quantification of message within the RNAseq data (NCBI Genome Expression Omnibus public database Series accession no. GSE150805). Similarly, we also did not see differences in previously activated/differentiated cells. While the proportion of cells expressing these receptors remained similar, it may still be possible that subtle differences in expression levels during the activation process may explain these biological effects, or there may be an unknown receptor(s)/co-receptor(s) that is required for the action of these cytokines, or some critical downstream molecule that may be different between naïve and memory cells. Future studies will be important in dissecting this biology.

In the same vein, there are interesting biologic implications of this preferential action on naïve CD4⁺ T cells. It is tempting to speculate that IL-17 cytokines may have an important role in microenvironments where T-cell priming is ongoing. For example, in the host's response to infection, the presence of IL-17 in these milieus (presumably coming from either pre-differentiated T-cells or other cell types responding to the same antigenic stimuli) would protect naïve T-cells from suppression and allow them to get primed in response to the ongoing infection. One can envision an opposite and detrimental scenario where autoantigen-specific T-cells are afforded similar immune resistance in the context of ongoing IL-17 production. These hypotheses can be addressed in future studies using appropriate modeling.

Amongst the cytokines tested, the IL-17AF heterodimer has the greatest effect. The combination of IL-17A and IL-17F can mimic the strong functional changes induced by IL-17AF [11]. However, it remains to be seen whether that combination also replicates the genetic program induced by the heterodimer. Several lines of inquiry further support the observation of partially overlapping functions. Mutations in IL-17RA and IL-17F have been linked to chronic

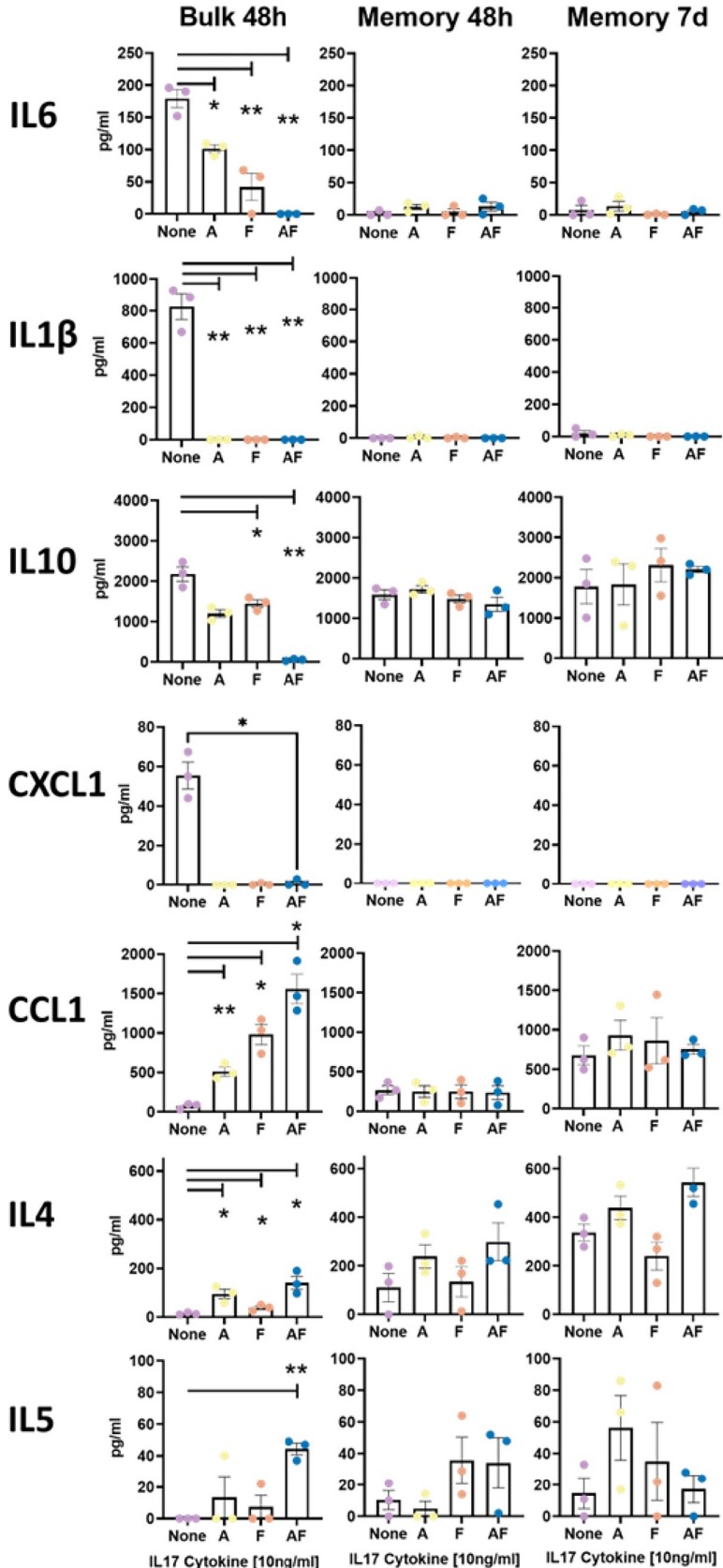

**Fig 7. ELISA measurements of selected cytokines and chemokines validate RNAseq analysis with significant differences between CD4+CD25- bulk T-cells at 48 hours but not between CD4+CD25-45RO+ memory T-cells at 48 hours or 7 days.** ELISAs were performed on supernatants frozen at -80 from cellular cultures of 48 hour bulk CD4+25- T-cell, 48-hour and 7-day memory CD4+25-45RO+ T-cells that were sent for RNA sequencing. N = 3. Data are +/- SEM. Statistical analysis is by paired Student t test *p<0.05, ** p<0.005.

mucocutaneous candidiasis disease and predisposition to *Staphylococcus aureus* infection [16]. Interestingly, the cases identified showed a difference with inheritance patterns, with IL-17RA mutations being autosomal recessive and IL-17F as autosomal dominant [16]. Differences in expression patterns have been noted in disease processes as well. For example, psoriatic lesions appear to have upregulation of both IL-17A and IL-17F [17], while upregulation of IL-17A, but not IL-17F have been noted in multiple sclerosis brain biopsy [6]. Conversely, IL-17F protein and mRNA seem more closely linked to the severity of asthma than IL-17A [18]. The common and divergent signaling produced by IL-17 ligands is unclear, especially with new observations of functions on CD4+ T-cells themselves.

Th17 cells and their cytokines are of immense interest in various clinical settings. For example, secukinumab, an agent targeting IL-17A, is an FDA-approved therapy for psoriasis, psoriatic arthritis, and ankylosing spondylitis [19–21]. However, this agent has not shown promising results in other autoimmune disease settings [22–24]. These discrepancies may be partially explained by differential functions of IL-17A vs IL-17F vs IL-17AF. Our data suggest that agents that interfere with the IL-17AF heterodimer or a combination of IL-17A and IL-17F may hold greater promise in some of these disease settings. Alternatively, a common downstream effector molecule may prove to be a better target for intervention. Thus, our studies provide a roadmap related to IL-17-mediated consequences on CD4+ T-cells that can be used to formulate immunotherapeutic intervention strategies and to understand the effects of anti-IL-17 therapy in specific diseases.

In summary, our findings provide novel insights into IL-17 function that may have implications in understanding the fundamental biology of these cytokines and inform the design of future interventional strategies.

## Supporting information

**S1 Fig. Magnetic bead separation sort purity in both positively and negatively sorted fractions.** *Ex vivo* CD4+25- bulk cells were magnetically sorted with either CD45RO+ or CD45RA+ Miltenyi positive selection beads. Mean purity of the sorts is 93.62% +/- 1.13; N = 5.
(TIFF)

**S2 Fig. Distribution of memory and naïve components of *ex vivo* CD4+CD25- T-cell sorts.** The bulk *ex vivo* CD4+25- T cell populations were made up of ~57–58% memory and ~42–43% naïve T-cells; N = 5.
(TIFF)

## Acknowledgments

The authors would like to thank Drs. Sushmita Sinha, Pranav Renavikar, Alexander Boyden, and Ashutosh Mangalam for helpful discussions and advice.

## Author Contributions

**Conceptualization:** Michael P. Crawford, Nicholas Borcherding, Nitin J. Karandikar.

**Data curation:** Nicholas Borcherding.

**Formal analysis:** Michael P. Crawford.

**Funding acquisition:** Nitin J. Karandikar.

**Investigation:** Nicholas Borcherding.

**Project administration:** Nitin J. Karandikar.

**Supervision:** Nitin J. Karandikar.

**Writing – original draft:** Michael P. Crawford, Nicholas Borcherding.

**Writing – review & editing:** Michael P. Crawford, Nicholas Borcherding, Nitin J. Karandikar.

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
