## [Decision Letter · Decision Letter 0]

12 Apr 2023

PONE-D-23-05159IL-17 cytokines preferentially act on naïve CD4+ T cells with the IL-17AF heterodimer inducing the greatest functional changesPLOS ONE

Dear Dr. Karandikar,

Thank you for submitting your manuscript to PLOS ONE. After careful consideration, we feel that it has merit but does not fully meet PLOS ONE’s publication criteria as it currently stands. Therefore, we invite you to submit a revised version of the manuscript that addresses the points raised during the review process.

We look forward to receiving your revised manuscript.

Kind regards,

Prof. Pierre Bobé

Academic Editor

PLOS ONE

Journal Requirements:

Reviewers' comments:

Reviewer's Responses to Questions

**Comments to the Author**

1. Is the manuscript technically sound, and do the data support the conclusions?

Reviewer #1: Yes

2. Has the statistical analysis been performed appropriately and rigorously? 

Reviewer #1: Yes

3. Have the authors made all data underlying the findings in their manuscript fully available?

Reviewer #1: Yes

4. Is the manuscript presented in an intelligible fashion and written in standard English?

Reviewer #1: Yes

5. Review Comments to the Author

Reviewer #1: The authors highlighted differential effect of IL-17 stimulation on naïve or memory CD4 T cells, as well as IL-17AF efficiency in promoting suppression-resistance of naïve CD4 T cells. The experiments have been conducted in a very satisfying way and the data therefore strongly support the authors’ conclusions.

Some minor modifications still must be made:

Nomenclature should be harmonized throughout the manuscript, e.g. IL17 receptor (Abstract line 37), anti-IL4 (Materials line 104), all CD4+ should be replaced by CD4+.

Similarly, some minor typo should be corrected, e.g. AlexaFlour (Materials, line 130)

Reviewer suggestions:

In figure 2D, it could be informative to add on the graph the statistical comparison of the proliferation percentage between Memory and Naïve for each tested ratio.

The same apply for Figure 3D, in which statistical comparison between “Same Well” and “Individually” could be informative.

For those two figures, highlighting the lack of statistical difference between the condition could reinforce the authors’ findings.

6. PLOS authors have the option to publish the peer review history of their article (what does this mean?). If published, this will include your full peer review and any attached files.

Reviewer #1: No

---

## [Author Response · Author response to Decision Letter 0]

14 Apr 2023

Re: Amended Manuscript PONE-D-23-05159

We would like to thank the editors and the reviewers for the constructive review of our manuscript and appreciate the opportunity to address your comments in our amended re-submission.

Here are point-by-point responses to the comments:

1. We note that you have included the phrase “data not shown” in your manuscript. Unfortunately, this does not meet our data sharing requirements. PLOS does not permit references to inaccessible data. We require that authors provide all relevant data within the paper, Supporting Information files, or in an acceptable, public repository. Please add a citation to support this phrase or upload the data that corresponds with these findings to a stable repository (such as Figshare or Dryad) and provide and URLs, DOIs, or accession numbers that may be used to access these data. Or, if the data are not a core part of the research being presented in your study, we ask that you remove the phrase that refers to these data.

We thank the editors for this comment/reminder. At one spot in the manuscript, we have removed the phrase completely as the data are not a core part of this study. At the other two places, we have replaced the “data not shown” phrase with the GEO public database information as these data are already shared there.

2. Reviewer #1: The authors highlighted differential effect of IL-17 stimulation on naïve or memory CD4 T cells, as well as IL-17AF efficiency in promoting suppression-resistance of naïve CD4 T cells. The experiments have been conducted in a very satisfying way and the data therefore strongly support the authors’ conclusions.

Some minor modifications still must be made:

Nomenclature should be harmonized throughout the manuscript, e.g. IL17 receptor (Abstract line 37), anti-IL4 (Materials line 104), all CD4+ should be replaced by CD4+.

Similarly, some minor typo should be corrected, e.g. AlexaFlour (Materials, line 130)

We thank the reviewer for these catches. We have now consistently added the hyphen when referring to the cytokine protein or receptor. Similarly, we have now superscripted the “+” and “-“ (positive and negative) references for CD4 as well as other markers (CD25, CD45RO, etc.). We have corrected the typo in line 130 as well.

3. Reviewer suggestions:

In figure 2D, it could be informative to add on the graph the statistical comparison of the proliferation percentage between Memory and Naïve for each tested ratio.

The same apply for Figure 3D, in which statistical comparison between “Same Well” and “Individually” could be informative.

For those two figures, highlighting the lack of statistical difference between the condition could reinforce the authors’ findings.

We thank the reviewers for these suggestions. We have conducted these statistical comparisons. To avoid clutter in the figures, we have only shown the statistically significant differences with asterisks. However, as suggested by the reviewers, we have now added a description about the non-significant comparisons within the text of the manuscript to highlight them and to reinforce our findings.

We hope that the modifications in the manuscript have addressed the reviewers’ and editors’ feedback and look forward to its publication in the PLOS ONE.

---

## [Editor Report · Decision Letter 1]

18 Apr 2023

IL-17 cytokines preferentially act on naïve CD4+ T cells with the IL-17AF heterodimer inducing the greatest functional changes

PONE-D-23-05159R1

Dear Dr. Karandikar,

We’re pleased to inform you that your manuscript has been judged scientifically suitable for publication and will be formally accepted for publication once it meets all outstanding technical requirements.

Kind regards,

Prof. Pierre Bobé

Academic Editor

PLOS ONE
---

## [Editor Report · Acceptance letter]

20 Apr 2023

PONE-D-23-05159R1 

IL-17 cytokines preferentially act on naïve CD4^+^ T cells with the IL-17AF heterodimer inducing the greatest functional changes 

Dear Dr. Karandikar:

I'm pleased to inform you that your manuscript has been deemed suitable for publication in PLOS ONE. Congratulations! Your manuscript is now with our production department. 

Kind regards, 

on behalf of

Prof Pierre Bobé 

Academic Editor

PLOS ONE